# A host enzyme reduces metabolic dysfunction-associated steatotic liver disease (MASLD) by inactivating intestinal lipopolysaccharide

Zhiyan Wang[1], Nore Ojogun[2], Yiling Liu[1], Lu Gan[1], Zeling Xiao[1], Jintao Feng[1], Wei Jiang[3], Yeying Chen[1], Benkun Zou[4], ChengYun Yu[1], Changshun Li[1], Asha Ashuo[5], Xiaobo Li[6], Mingsheng Fu[7], Jian Wu[5†], Yiwei Chu[1], Robert S Munford[2,8]*, Mingfang Lu[1,9,10]*

[1]Department of Immunology, School of Basic Medical Sciences, Department of Trauma-Emergency & Critical Care Medicine, Shanghai Fifth People's Hospital, Fudan University, Shanghai, China; [2]Infectious Disease Division, Department of Internal Medicine, University of Texas Southwestern Medical Center, Dallas, United States; [3]Department of Rheumatology and Immunology, the Affiliated Hospital of Guizhou Medical University, Guizhou, China; [4]BeiGene Institute, BeiGene (Shanghai) Research and Development Co., Ltd, Shanghai, China; [5]Department of Medical Microbiology and Parasitology, MOE/NHC/CAMS Key Laboratory of Medical Molecular Virology, School of Basic Medical Sciences, Fudan University, Shanghai, China; [6]Department of Physiology and Pathophysiology, School of Basic Medical Sciences, Fudan University, Shanghai, China; [7]Department of Gastroenterology, Shanghai Fifth People's Hospital, Fudan University, Shanghai, China; [8]Antibacterial Host Defense Unit, Laboratory of Clinical Immunology and Microbiology, National Institute of Allergy and Infectious Diseases (NIAID), National Institutes of Health (NIH), Bethesda, United States; [9]MOE Innovative Center for New Drug Development of Immune Inflammatory Diseases, Fudan University, Shanghai, China; [10]Shanghai Sci-Tech Inno Center for Infection and Immunity, Shanghai, China

*For correspondence:
munfordrs@niaid.nih.gov (RSM);
mingfanglu@fudan.edu.cn (ML)

†Deceased

## eLife Assessment

This **important** study highlights the key role of the gut–liver axis mediated by LPS in causing hepatic steatosis. The authors provide **solid** evidence, in vivo, in vitro, and in silico, for the role of acyloxy-acyl hydrolase in mediating this effect using KO mice subjected to MASD-inducing diets. The findings are significant for the liver research community and others interested in the gut–liver axis.

**Abstract** The incidence of metabolic dysfunction-associated steatotic liver disease (MASLD) has been increasing worldwide. Since gut-derived bacterial lipopolysaccharides (LPS) can travel via the portal vein to the liver and play an important role in producing hepatic pathology, it seemed possible that (1) LPS stimulates hepatic cells to accumulate lipid, and (2) inactivating LPS can be preventive. Acyloxyacyl hydrolase (AOAH), the eukaryotic lipase that inactivates LPS and oxidized phospholipids, is produced in the intestine, liver, and other organs. We fed mice either normal chow or a high-fat diet for 28 weeks and found that *Aoah⁻/⁻* mice accumulated more hepatic lipid than did *Aoah⁺/⁺* mice. In young mice, before increased hepatic fat accumulation was observed,

*Aoah^{-/-}* mouse livers increased their abundance of sterol regulatory element-binding protein 1, and the expression of its target genes that promote fatty acid synthesis. *Aoah^{-/-}* mice also increased hepatic expression of *Cd36* and *Fabp3*, which mediate fatty acid uptake, and decreased expression of fatty acid-oxidation-related genes *Acot2* and *Ppara*. Our results provide evidence that increasing AOAH abundance in the gut, bloodstream, and/or liver may be an effective strategy for preventing or treating MASLD.

## Introduction

Metabolic dysfunction-associated steatotic liver disease (MASLD) is a common human affliction. Its global prevalence, currently about 25%, has been increasing (*Diehl and Day, 2017*; *Fan et al., 2017*; *Younossi et al., 2018*). MASLD may progress from nonalcoholic fatty liver to nonalcoholic steatohepatitis, cirrhosis, and even hepatic cancer (*Friedman et al., 2018*; *Sheka et al., 2020*). Multiple factors may contribute to its pathogenesis (*Friedman et al., 2018*) prominent among these are the lipopolysaccharides (LPS, endotoxins) produced by many of the Gram-negative bacteria that inhabit the intestine. Gut-derived LPS may translocate into the portal venous system and traffic to the liver, triggering or exacerbating hepatic inflammation (*Albillos et al., 2020*; *Carpino et al., 2020*; *Han et al., 2021*; *Kazankov et al., 2019*; *Leung et al., 2016*; *Munford, 1978*; *Wang et al., 2022*).

The LPS molecules that contribute to MASLD pathogenesis are able to stimulate host cells because their lipid A structure is recognized by MD-2/TLR4 receptors (*Ye et al., 2012*). Most γ-*Proteobacteria* such as *Escherichia coli* produce stimulatory hexaacyl LPS (with six acyl chains) while *Bacteroidetes* produce non-stimulatory LPS that has four or five acyl chains (*Anhê et al., 2021*; *d'Hennezel et al., 2017*). MASLD is often associated with intestinal dysbiosis produced by increased abundance of γ-*Proteobacteria*, which leads to tissue inflammation and increased intestinal permeability that allows even more gut-derived LPS to reach the liver (*Albillos et al., 2020*; *Aron-Wisnewsky et al., 2020a*; *Aron-Wisnewsky et al., 2020b*; *Mouries et al., 2019*; *Rodrigues et al., 2024*). Although gut-derived LPS is known to induce hepatic inflammation and exacerbate MASLD, how LPS influences hepatocyte fatty acid metabolism before MASLD develops has not been well understood.

Acyloxyacyl hydrolase (AOAH) is a highly conserved animal lipase that is mainly expressed in macrophages, monocytes, neutrophils, microglia, dendritic cells, NK cells, and ILC1 cells (*Munford et al., 2020*). It can inactivate Gram-negative bacterial LPSs by releasing two of the six fatty acyl chains present in the lipid A moiety (*Figure 1A*). It also can deacylate/inactivate oxidized phospholipids and lysophospholipids, molecules that are also known to contribute to MASLD (*Sun et al., 2020*; *Zou et al., 2021*). We previously reported that AOAH is expressed by gut macrophages and dendritic cells and can inactivate bioactive LPS in feces (*Cheng et al., 2023*; *Janelsins et al., 2014*; *Qian et al., 2018*). AOAH also inactivates LPS in the liver, diminishing and shortening hepatic inflammation induced by bloodborne LPS (*Ojogun et al., 2009*; *Shao et al., 2011*; *Shao et al., 2007*). Han et al. found that intestine-derived LPS can bind high-density lipoprotein 3 (HDL_3) and be inactivated by AOAH as it traffics from the jejunum to the liver via the portal vein (*Han et al., 2021*). The enzyme's ability to prevent MASLD by inactivating gut-derived LPS had not been tested (*Ojogun, 2008*).

In this study, we found that when mice were fed either normal chow (NC) or a high-fat diet (HFD), AOAH reduced LPS-induced lipid accumulation in the liver, probably by decreasing the expression and activation of sterol regulatory element-binding protein 1 (SREBP1) (also called sterol regulatory element-binding transcription factor 1 [SREBF1]), an important transcription factor that promotes fatty acid synthesis (*Horton et al., 2002*; *Shimano and Sato, 2017*). AOAH also reduced the expression of *Cd36* and *Fabp3*, fatty acid uptake-related genes, and increased that of fatty acid oxidation-related genes (*Acot2* and *Ppara*). In addition, AOAH reduced hepatic inflammation and minimized tissue damage. Our results suggest that AOAH plays a regulatory role in ameliorating MASLD and that measures that increase AOAH abundance in the intestine, liver, and/or bloodstream may help prevent this common disease.

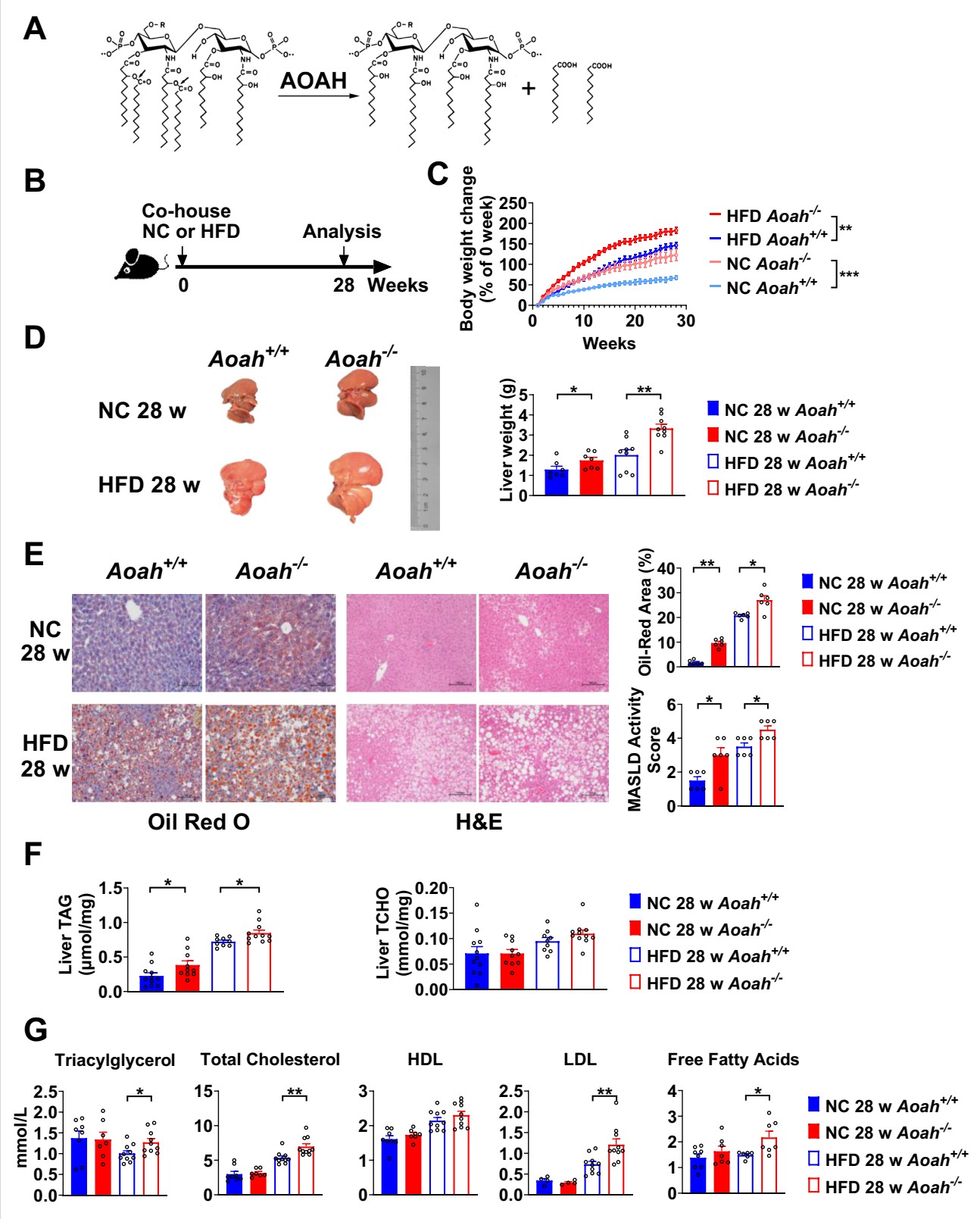

**Figure 1.** Acyloxyacyl hydrolase (AOAH) reduces hepatic lipid accumulation. (**A**) AOAH cleaves the two piggyback fatty acyl chains from the lipid A moiety, inactivating lipopolysaccharides (LPS). The arrows indicate the cleavage sites. (**B**) Co-housed *Aoah*$^{+/+}$ and *Aoah*$^{-/-}$ mice were fed either a normal diet (NC) or a high-fat diet plus high fructose (23.1 g/l) and glucose (18.9 g/l) in their drinking water (HFD) for 28 weeks. (**C**) Body weight was measured weekly for 28 weeks. Data were combined from four experiments. n=12–17. (**D**) Representative images of livers at 28 weeks are shown, and the liver

*Figure 1 continued on next page*

*Figure 1 continued*

weight was measured. n=7–9, each symbol represents one mouse. (**E**) Mouse livers were fixed, sectioned, and stained with Oil Red O or H&E. In Oil Red O-stained sections, the percentage of area occupied by the lipid droplets was quantified using ImageJ. H&E staining results were semi-quantitatively scored for disease severity. Data were combined from three experiments, n=6. Scare bars = 100 µm. (**F**) Triacylglycerol (TAG) and total cholesterol (TCHO) were measured in liver homogenates. Data were combined from at three experiments, n=9–11, each symbol represents one mouse. (**G**) The serum concentrations of triglyceride, TCHO, high-density lipoprotein (HDL), low-density lipoprotein (LDL), and free fatty acids were measured. Data were combined from three experiments. n=6–10, each symbol represents one mouse. (**C–G**) Mann–Whitney test and two-way ANOVA were used. *p<0.05; **p<0.01; ***p<0.001.

## Results

### AOAH reduces hepatic lipid accumulation

To find out if AOAH prevents MASLD, we fed co-housed $Aoah^{+/+}$ and $Aoah^{-/-}$ mice NC or a HFD plus fructose and glucose in the drinking water, for 28 weeks (*Figure 1B*; *Liu et al., 2018*). $Aoah^{-/-}$ mice fed either NC or HFD gained more weight than did $Aoah^{+/+}$ control mice (*Figure 1C*). When they were fed either NC or HFD, the livers of $Aoah^{-/-}$ mice were heavier than those of $Aoah^{+/+}$ control mice (*Figure 1D*). Histological examination and Oil Red O staining revealed that $Aoah^{-/-}$ mouse livers accumulated more lipid droplets than did the livers of $Aoah^{+/+}$ mice (*Figure 1E*). When we scored MASLD severity based on steatosis, hepatocyte ballooning degeneration, and inflammation, $Aoah^{-/-}$ mice developed more severe MASLD than did $Aoah^{+/+}$ mice whether the mice were fed NC or HFD (*Sheka et al., 2020*; *Figure 1E*). When the mice were fed either NC or HFD, $Aoah^{-/-}$ mouse livers contained more triacylglycerol (TAG) than did $Aoah^{+/+}$ mouse livers, while livers from both mouse strains contained a similar amount of total cholesterol (TCHO, *Figure 1F*). When the mice were fed the HFD, $Aoah^{-/-}$ mice had higher serum levels of TAG, TCHO, low-density lipoprotein (LDL), and free fatty acids than did $Aoah^{+/+}$ mice (*Figure 1G*). Collectively, these findings were evidence that AOAH reduced hepatic TAG accumulation when mice were fed either NC or HFD.

### AOAH prevents hepatic inflammation and tissue injury when mice are fed HFD

$Aoah^{-/-}$ mice fed HFD had significantly higher serum alanine aminotransferase (ALT) and aspartate aminotransferase (AST) levels than did control $Aoah^{+/+}$ mice, suggesting that $Aoah^{-/-}$ mice experienced more severe liver inflammation and tissue damage (*Figure 2A*). To assess liver inflammation, we measured pro- and anti-inflammatory cytokine expression. When mice were fed the HFD, $Aoah^{-/-}$ mouse livers produced more pro-inflammatory *Il6*, *Ifng,* and anti-inflammatory *Il10* mRNA than did $Aoah^{+/+}$ mouse livers, suggesting greater inflammation (*Figure 2B*). We also found that $Aoah^{-/-}$ livers had more *Timp1* (a pro-fibrosis gene) mRNA and less *Mmp2* (an anti-fibrosis gene) mRNA than did $Aoah^{+/+}$ mouse livers (*Kisseleva and Brenner, 2021*), indicating that $Aoah^{-/-}$ mouse livers may be developing more severe fibrosis, although we did not detect fibrosis with Masson staining (*Figure 2C*). We analyzed the myeloid cells in the liver (*Daemen et al., 2021*) and found that when the mice were fed the HFD, $Aoah^{-/-}$ mouse livers contained more neutrophils, monocytes, and lipid-associated macrophages (hepatic LAMs) (*Remmerie et al., 2020*; *Su, 2002*) than did $Aoah^{+/+}$ mouse livers (*Figure 2D and E*). Collectively, when the mice were fed an HFD, the livers of $Aoah^{-/-}$ mice developed significantly greater inflammatory responses and tissue damage.

### AOAH reduces hepatic LPS levels and the expression of fatty acid synthesis genes

We found previously that AOAH was mainly expressed in Kupffer cells in the liver (*Shao et al., 2007*). To confirm the previous results, we first consulted the single-cell RNA-seq analysis reported by *Remmerie et al., 2020*, who found that AOAH is expressed in Kupffer cells, monocytes, monocyte-derived cells, NK (circulating NK), and ILC1 (tissue resident NK) cells in mouse livers (*Remmerie et al., 2020*; *Figure 3A*). We used flow cytometry to sort mouse Kupffer cells (CD45[+]NK1.1[-] F4/80[hi]CD11b[mid]), monocytes (CD45[+]NK1.1[-] F4/80[mid]CD11b[hi]), NK cells (CD45[+]SSC[lo] NK1.1[+]) (including circulating NK and resident ILC1), and purified the hepatocytes. Using qPCR analysis, we found that AOAH mRNA was present in Kupffer cells, monocytes, and NK cells but not in hepatocytes, in keeping with previous findings (*Shao et al., 2007*; *Figure 3B*). Western blot analysisanalysis confirmed that $Aoah^{+/+}$ mouse

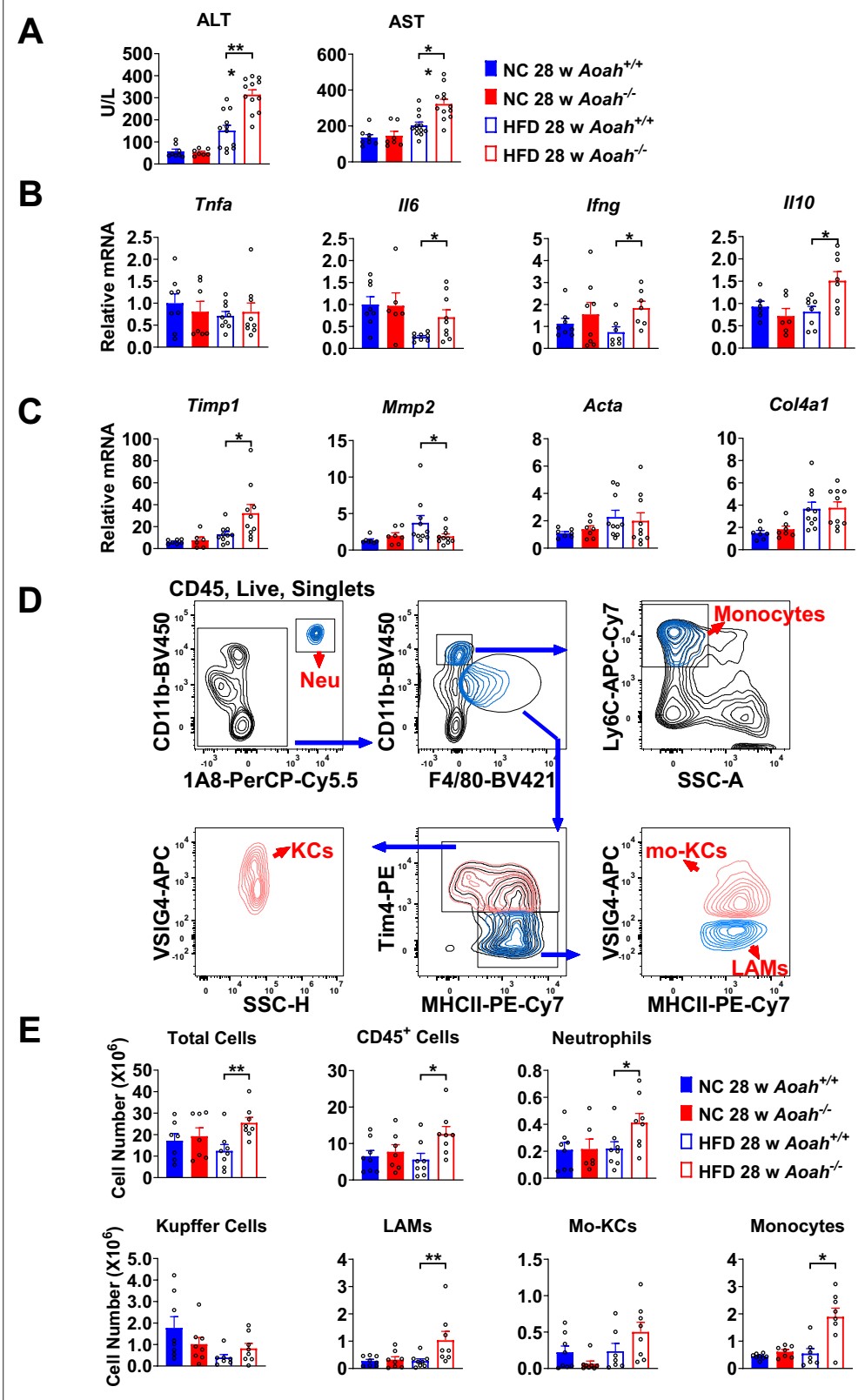

**Figure 2.** Acyloxyacyl hydrolase (AOAH) prevents hepatic inflammation and tissue injury when mice are fed high-fat diet (HFD). (**A**) Serum alanine aminotransferase (ALT) and aspartate aminotransferase (AST) were measured at 28 weeks. Data were combined from three experiments, n=7–12. (**B, C**) Inflammation-related *Tnfa, Il6, Ifng, Il10* mRNA and fibrosis-related *Timp1, Mmp2, Acta, Col4a1* mRNA were measured in livers at 28 weeks. Data were

*Figure 2 continued on next page*

*Figure 2 continued*

combined from three experiments, n=6–10. (**D**) Gating strategy to identify hepatic neutrophils, monocytes, Kupffer cells, lipid-associated macrophages (LAM), and monocyte-derived Kupffer cells (Mo-KC) subsets. (**E**) The myeloid cell numbers in *Aoah*$^{+/+}$ and *Aoah*$^{-/-}$ livers were calculated using FACS analysis. Data were combined from three experiments, n=6–8. Each symbol represents one mouse. Mann–Whitney test was used. *p<0.05; **p<0.01.

livers but not *Aoah*$^{-/-}$ mouse livers had AOAH protein (*Figure 3C*). When *Aoah*$^{+/+}$ mice aged or had an HFD, their hepatic AOAH expression increased (*Figure 3D*). We also studied the expression of AOAH in human liver cells by analyzing the single-cell RNA-seq data obtained by *Ramachandran et al., 2019*; AOAH was expressed in macrophages, monocytes, resident and circulating NK cells, and some T cells (*Figure 3E*). AOAH expression increased in liver macrophages and monocytes in MASLD patients (*Figure 3E*). As gut-derived LPS can be transported via the portal vein into the liver (*Han et al., 2021*), we hypothesized that AOAH prevents hepatic inflammation and fat accumulation by inactivating LPS in the gut, portal vein, and liver. We found that *Aoah*$^{-/-}$ mouse feces, liver, and plasma had higher bioactive LPS levels when *Aoah*$^{-/-}$ and *Aoah*$^{+/+}$ mice were fed either NC or HFD (*Figure 3F*). HFD increased gut permeability, but there was no permeability difference between *Aoah*$^{+/+}$ and *Aoah*$^{-/-}$ mice that were fed either NC or HFD for 28 weeks, suggesting that the increased hepatic LPS levels in *Aoah*$^{-/-}$ mouse livers were mainly caused by failure to inactivate LPS (*Figure 3G*). After the mice were fed HFD for 28 weeks, *Aoah*$^{-/-}$ mouse livers increased fatty acid uptake gene *Fabp3* mRNA and fatty acid synthesis gene *Fasn* mRNA, changes that may have contributed to lipid accumulation (*Figure 3H*). Thus, AOAH may prevent hepatic lipid accumulation by diminishing bioactive LPS in the liver.

## AOAH can regulate the expression of hepatic fatty acid metabolism genes

We found that AOAH reduced hepatic lipid accumulation when mice were about 8 months old if they were fed either NC or HFD. To investigate the mechanism, we analyzed the livers of young mice. We co-housed 3–4-week-old *Aoah*$^{+/+}$ and *Aoah*$^{-/-}$ mice for 3–4 more weeks before removing their livers for RNA-seq analysis. The expression of several fatty acid biosynthesis genes, such as *Acacb* (acetyl-coenzyme A carboxylase beta), *Acss2* (acetyl-CoA synthetase 2), *Pcx* (pyruvate carboxylase), *Acly* (ATP citrate lyase), *Fasn* (fatty acid synthase), and *Scd1* (stearoyl-coenzyme A desaturase 1), was significantly increased in *Aoah*$^{-/-}$ mouse livers (*Figure 4A*). When we used gene set enrichment analysis (GSEA) to analyze deferentially expressed genes, we found that the fatty acid biosynthesis pathway was enriched (*Figure 4B*). We then did qPCR and confirmed the increases in mRNAs for FA biosynthesis genes described in *Figure 4A* as well as for *Acaca* (acetyl-coenzyme A carboxylase alpha, encoding ACC1, the first and key enzyme on FA synthesis pathway) in *Aoah*$^{-/-}$ mouse livers compared with *Aoah*$^{+/+}$ mouse livers (*Figure 4C*). In addition, mRNAs for enzymes involved in fatty acid oxidation (*Acot2* [acyl-CoA thioesterase 2] and *Ppara* [peroxisome proliferator-activated receptor α]) (*Bougarne et al., 2018*; *Moffat et al., 2014*) decreased (*Figure 4D*), while *Cd36* (fatty acid uptake; *Chen et al., 2022*) mRNA levels increased in *Aoah*$^{-/-}$ mouse livers (*Figure 4E*). We confirmed that FASN and SCD1 protein levels also increased in *Aoah*$^{-/-}$ mouse livers (*Figure 4F*). We found previously that LPS and other TLR agonists increase lipid accumulation in cultured macrophages by increasing expression of *Acsl1* (acyl-CoA synthetase long-chain family member 1) and *Dgat2* (diacylglycerol O-acyltransferase 2) and by reducing the production of *Pnpla2* (Patatin-like phospholipase domain containing 2, *Atgl*) (*Huang et al., 2014*), yet the livers of *Aoah*$^{+/+}$ and *Aoah*$^{-/-}$ mice had similar levels of *Acsl1*, *Dgat2,* and *Pnpla2* mRNA, suggesting that AOAH does not regulate hepatic TAG metabolism (*Figure 4—figure supplement 1*). These results suggest that AOAH reduces liver fat accumulation by diminishing the expression of fatty acid synthesis and uptake genes and increasing that of fatty acid oxidation genes.

## AOAH reduces hepatic SREBP1

As *Acaca*, *Fasn,* and *Scd1* are all target genes for sterol regulatory element-binding protein 1 (SREBP1, encoded by *Srebf1* gene), a key transcription factor for fatty acid biosynthesis, we next analyzed SREBP1 expression in the liver (*Horton et al., 2002*; *Shimano and Sato, 2017*; *Yokoyama et al., 1993*). There are two isoforms of SREBP1, SREBP1a, and SREBP1c, and the liver predominantly expresses SREBP1c (*Horton et al., 2002*). The abundance of *Srebf1a* and *Srebf1c* mRNA increased in

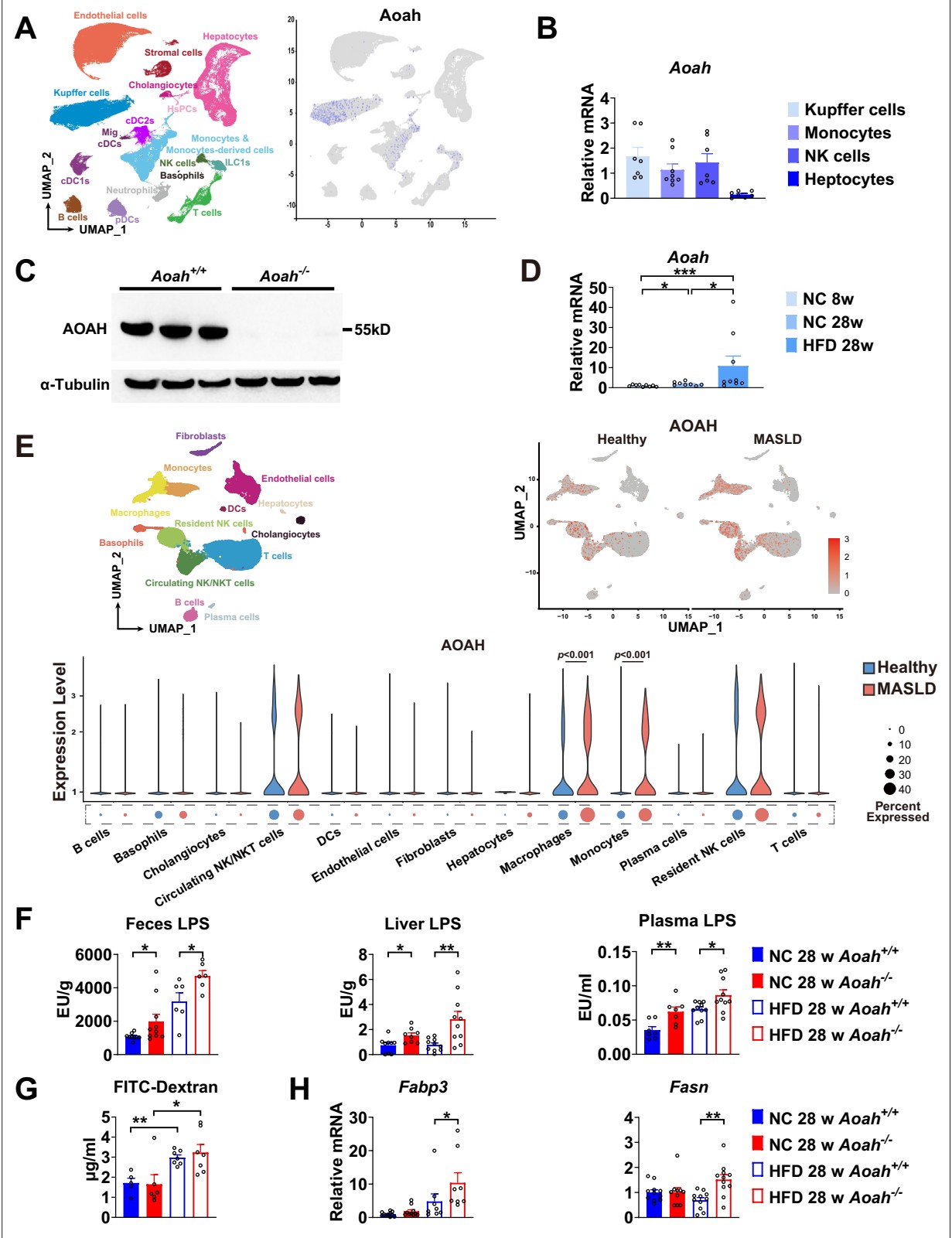

**Figure 3.** Acyloxyacyl hydrolase (AOAH) reduces hepatic lipopolysaccharides (LPS) levels and the expression of fatty acid uptake and synthesis genes. (**A**) AOAH expression in mouse hepatic cells based on single-cell analysis by *Remmerie et al., 2020* is shown, n=4. (**B**) Kupffer cells, monocytes, and NK cells were sorted using flow cytometry and hepatocytes were purified from 6- to 8-week-old mice. AOAH mRNA was measured. Data were combined from three experiments, n=7–8, each symbol represents one mouse. (**C**) Using western blot analysis, we confirmed that AOAH protein was

*Figure 3 continued on next page*

*Figure 3 continued*
present in 6–8 week-old *Aoah*[+/+] mouse livers but not in *Aoah*[-/-] mouse liver homogenates. Similar results were obtained in two other experiments. (**D**) AOAH mRNA levels were measured in the livers of *Aoah*[+/+] mice fed with a normal chow for 8 and 28 weeks, and a high-fat diet for 28 weeks. Data were combined from 2 experiments, n=5–7, each symbol represents one mouse. (**E**) Single-cell RNA sequencing data from livers of MASLD patients and healthy controls by *Ramachandran et al., 2019*. n=5/group. (**F**) Heat-inactivated feces suspension, liver homogenates, and plasma from *Aoah*[+/+] and *Aoah*[-/-] mice were tested for TLR4-stimulating activity. Data were combined from at least three experiments, n=6–10, each symbol represents one mouse. (**G**) Gut permeability was measured. Mice were fasted for 18 h. Mice were orally gavaged with fluorescein isothiocyanate (FITC)-conjugated dextran and 4 h later, FITC fluorescence was measured in plasma. Data were combined from three experiments, n=5–7, each symbol represents one mouse. (**H**) At 28 weeks of normal diet (NC) or high-fat diet (HFD) feeding, liver *Fabp3* and *Fasn* mRNAs were measured. Data were combined from three experiments, n=8–11, each symbol represents one mouse. Mann–Whitney test was used. *p<0.05; **p<0.01.

The online version of this article includes the following source data for figure 3:

**Source data 1.** Original tiff files of western blots for *Figure 3C*.

**Source data 2.** Original tiff files containing uncropped western blots with labeling for *Figure 3C*.

the livers of young *Aoah*[-/-] mice (*Figure 5A*). SREBP1 is synthesized as a 125 kDa precursor (full length, flSREBP1) in the endoplasm reticulum, transferred to the Golgi apparatus, and cleaved sequentially by Site-1 protease and Site-2 protease to generate nuclear SREBP1 (nSREBP1, 68 kDa), which enters the nucleus and activates fatty acid biosynthesis gene transcription (*Horton et al., 2002*; *Shimano and Sato, 2017*). The livers mainly had the short-form (68 kDa) nSREBP1 protein, which was significantly more abundant in *Aoah*[-/-] mice (*Figure 5B*). We separated liver cytosol and nuclei and found that the short-form SREBP1 was mainly present in nuclei and that *Aoah*[-/-] mouse liver nuclei contained significantly more SREBP1 than did *Aoah*[+/+] mouse liver nuclei (*Figure 5C*).

When we analyzed the transcript profiles from co-housed 6–8-week-old *Aoah*[+/+] and *Aoah*[-/-] mouse livers, we noticed that the expression of inflammatory genes for serum amyloid A1 (*Saa1*), *Saa2*, and *Saa3* was significantly higher in *Aoah*[-/-] mouse livers (*Figure 5—figure supplement 1A*). We isolated hepatocytes from *Aoah*[+/+] and *Aoah*[-/-] mice and found that *Aoah*[-/-] mouse hepatocytes expressed increased levels of *Saa1*, *Saa2*, *Saa3*, and inflammation regulatory *Irak3* mRNA (*Figure 5—figure supplement 1B*). Purified hepatocytes from *Aoah*[-/-] mice had increased levels of *Acly*, *Acaca*, *Acacb*, *Fasn*, and *Cd36* mRNA (*Figure 5—figure supplement 1C and D*) and decreased levels of *Acot2* and *Ppara* mRNA (*Figure 5—figure supplement 1E*), changes that may contribute to lipid accumulation as mice grow older. Thus, in young mice, even before hepatic lipid accumulation can be observed, *Aoah*[-/-] mouse hepatocytes have altered expression of genes that may promote lipid storage.

We next used clodronate-liposomes to deplete Kupffer cells and found that clodronate-liposome treatment diminished liver *Aoah* mRNA, confirming that Kupffer cells are the major source of hepatic AOAH (*Shao et al., 2007*; *Figure 5D*). Notably, after clodronate liposome treatment, nSREBP1 levels increased in the liver significantly, resembling *Aoah*[-/-] mice (*Figure 5D*). AKT-mTOR1-p70 S6-kinase (S6K) activation induces SREBP1c processing in hepatocytes (*Jeon et al., 2023*; *Owen et al., 2012*; *Yecies et al., 2011*). Livers from clodronate-liposome-treated mice had increased AKT and mTOR activation (*Figure 5D*), suggesting that when gut-derived LPS cannot be inactivated by AOAH in the liver, bioactive LPS stimulates the mTOR pathway and induces SREBP1 activation.

## Excessive gut-derived LPS increases hepatic nSREBP1 and mTOR activation

To find out if excessive gut LPS increases liver LPS levels and promotes fatty acid synthesis gene expression, we orally gavaged *Aoah*[+/+] mice with LPS. We confirmed that orally gavaged (i.g.) LPS increased hepatic LPS levels in *Aoah*[+/+] mice (*Figure 6A*). Like *Aoah*[-/-] mice, *Aoah*[+/+] mice that received gavaged LPS had increased levels of *Srebf1a*, *Srebf1c*, *Pcx*, *Acaca*, *Acacb*, *Fasn*, *Scd1*, and *Cd36* mRNA in their livers (*Figure 6B and C*). LPS administered i.g. also increased nSREBP1 in *Aoah*[+/+] mouse livers (*Figure 6D*). Consistently, elevated AKT-mTOR-S6K activation was found in *Aoah*[-/-] mouse livers, and when we gave *Aoah*[+/+] mice i.g. LPS, hepatic AKT-mTOR-S6K activity increased (*Figure 6D*). To find out whether LPS can directly stimulate hepatocytes to induce SREBP1 activation, we isolated primary hepatocytes and found that LPS stimulated mTOR activation and nSREBP1 upregulation (*Figure 6E*). Adding purified Kupffer cells to the hepatocyte culture did not further increase SREBP1 activation, suggesting that LPS directly acts on hepatocytes (*Yu et al., 2021*), at least in vitro. Blocking mTOR activation using torin1 prevented LPS-induced nSREBP1 upregulation (*Figure 6E*), suggesting that

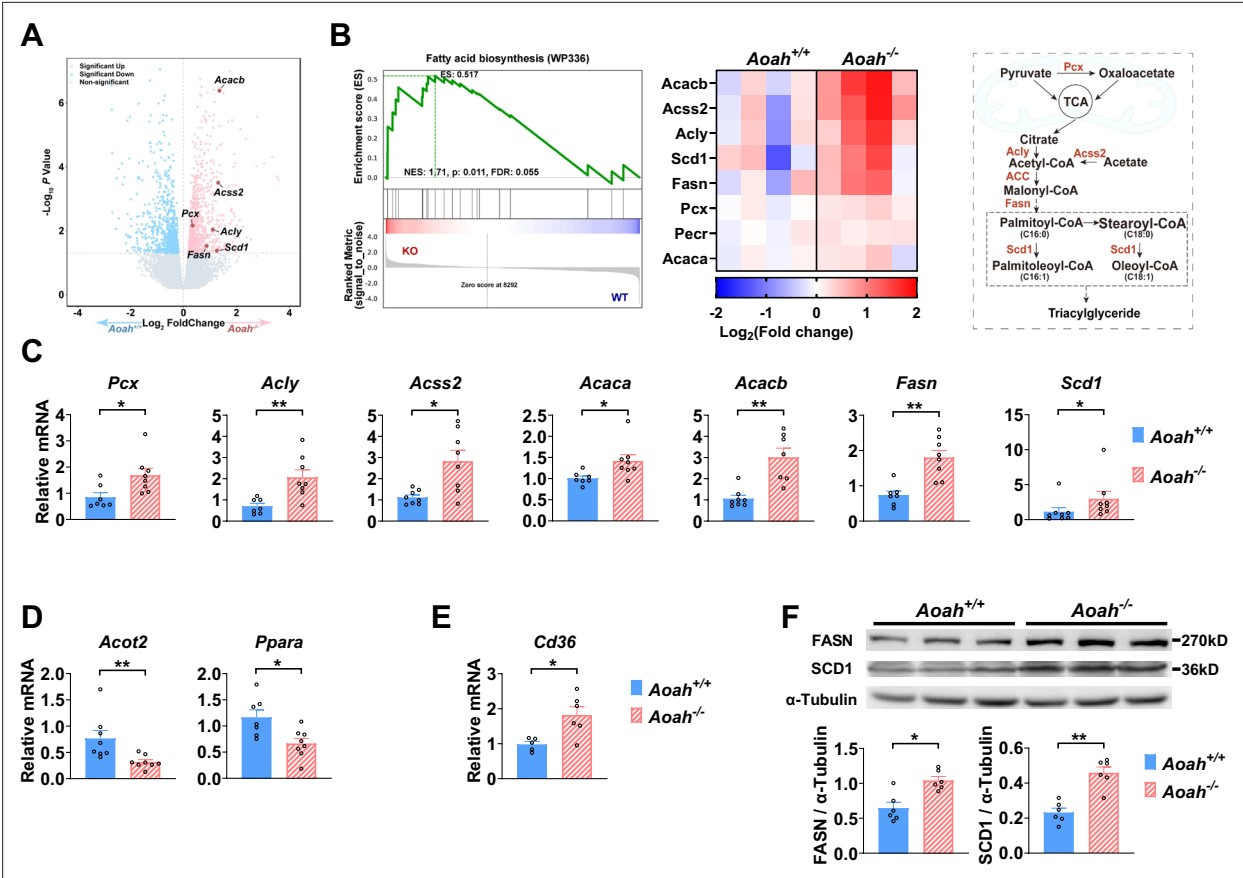

**Figure 4.** Acyloxyacyl hydrolase (AOAH) regulates the expression of hepatic fatty acid metabolism genes. (**A, B**) Co-housed 6–8-week-old (i.e., young) *Aoah*+/+ and *Aoah*-/- mouse livers were subjected to RNA-seq analysis. The differentially expressed genes are shown in the volcano plot (**A**). The fatty acid biosynthesis pathway was found enriched using gene set enrichment analysis (GSEA) (**B**, left panel). The fatty acid biosynthesis-associated gene expression levels of co-housed *Aoah*+/+ and *Aoah*-/- mouse livers are shown as a heatmap, n=4 (**B**, middle panel). The fatty acid biosynthesis pathway is shown. The enzymes marked red had increased expression in young *Aoah*-/- mouse livers (**B**, right panel). (**C–E**) The hepatic expression of fatty acid synthesis (**C**), oxidation (**D**), and uptake (**E**) genes was measured in co-housed 6–8-week-old *Aoah*+/+ and *Aoah*-/- mice. Data were combined from three experiments, n=5–8, each symbol represents one mouse. (**F**) Liver homogenates from co-housed 6–8-week-old *Aoah*+/+ and *Aoah*-/- mice were subjected to western analysis. FASN, SCD1, and α-tubulin protein levels were quantitated using ImageJ. Data were combined from two experiments, n=6, each symbol represents one mouse. Mann–Whitney test was used. *p<0.05; **p<0.01.

The online version of this article includes the following source data and figure supplement(s) for figure 4:

**Source data 1.** Original tiff files of western blots for *Figure 4F*.

**Source data 2.** Original tiff files containing uncropped western blots with labeling for *Figure 4F*.

**Figure supplement 1.** *Aoah*+/+ and *Aoah*-/- mouse livers express similar levels of triacylglycerol metabolism mRNA.

upon LPS stimulation SREBP1 activation depends upon the mTOR pathway. Collectively, these results confirm that excessive hepatic LPS derived from the *Aoah*-/- mouse intestine induces hepatocyte mTOR activity, which increases nSREBP1 abundance; AOAH prevents hepatic lipid accumulation by inactivating gut-derived LPS.

## Discussion

The intestine and liver are connected by the portal vein, enabling the transport of gut commensal-derived molecules, including Gram-negative bacterial LPS, to the liver (***Albillos et al., 2020***; ***Leung et al., 2016***). Much evidence suggests that gut-derived LPS induces hepatic inflammation and therefore exacerbates MASLD, especially when dysbiosis and intestinal barrier dysfunction have occurred (***An et al., 2022***; ***Aron-Wisnewsky et al., 2020a***; ***Aron-Wisnewsky et al., 2020b***; ***Leung et al., 2016***). As lipid accumulation in hepatocytes is considered to be the first hit, gut-derived LPS is usually

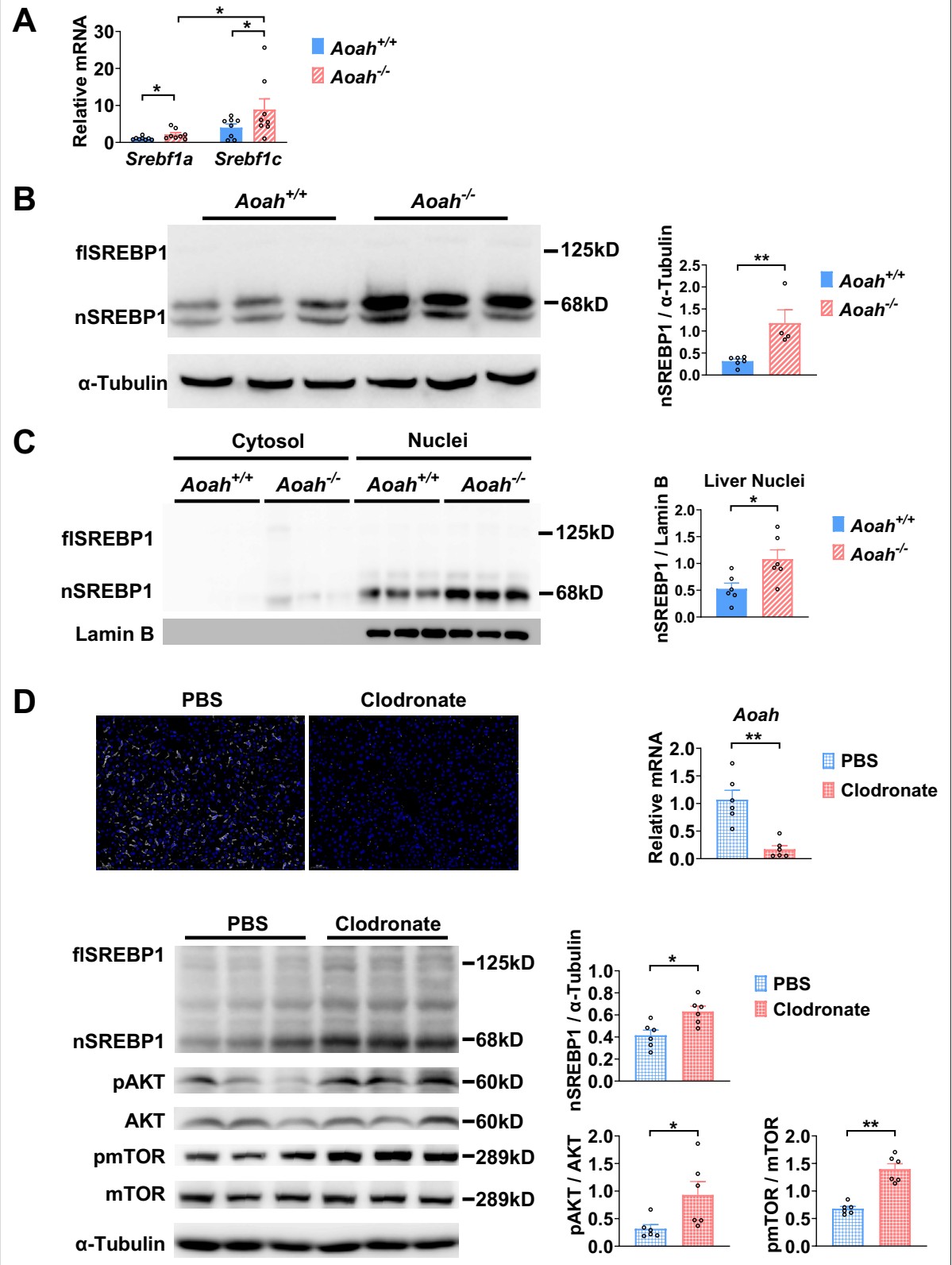

**Figure 5.** Acyloxyacyl hydrolase (AOAH) reduces hepatic SREBP1. (**A**) *Srebf1a* and *Srebf1c* gene mRNA was measured in livers from *Aoah*⁺/⁺ and *Aoah*⁻/⁻ mice. n=8, each symbol represents one mouse. (**B**) Liver homogenates from *Aoah*⁺/⁺ and *Aoah*⁻/⁻ mice were subjected to western analysis. SREBP1 protein levels were quantitated using ImageJ. flSREBP1 is full-length SREBP1, which is a precursor; nSREBP1is nuclear SREBP1, which is an active form. n=4–6, each symbol represents one mouse. (**C**) Liver cytosol and nuclei were separated from *Aoah*⁺/⁺ and *Aoah*⁻/⁻ mice and then subjected to western

*Figure 5 continued on next page*

*Figure 5 continued*

analysis. SREBP1 protein levels were quantitated using ImageJ. n=6, each symbol represents one mouse. (**D**) *Aoah*[+/+] mice were injected i.v. with 200 μl clodronate-liposomes or PBS-liposomes. After 2 days, livers were dissected, sectioned, and stained with anti-F4/80 antibody (white) and DAPI (blue). AOAH mRNA was measured in livers. SREBP1, pAKT, AKT, pmTOR, and mTOR were measured using western blotting. n=6, each symbol represents one mouse. Mann–Whitney test was used. *p<0.05; **p<0.01.

The online version of this article includes the following source data and figure supplement(s) for figure 5:

**Source data 1.** Original tiff files of western blots for *Figure 5B–D*.

**Source data 2.** Original tiff files containing uncropped western blots with labeling for *Figure 5B–D*.

**Figure supplement 1.** *Aoah*[-/-] mouse hepatocytes have altered expression of genes that may promote lipid storage.

thought to be the second hit in the pathogenesis of MASLD, mainly inducing inflammation (***An et al., 2022***), yet the possibility that LPS also has direct effects on hepatocyte lipid metabolism has received little attention. In previous studies, we found that when we co-housed *Aoah*[+/+] and *Aoah*[-/-] mice for three or more weeks, they had similar microbiota (***Qian et al., 2018***), yet we found significantly more LPS in *Aoah*[-/-] mouse feces and livers. In this study, we found that *Aoah*[-/-] mice accumulated more hepatic fat than did *Aoah*[+/+] mice when the mice were fed either NC or an HFD. *Aoah*[-/-] mouse livers also expressed more inflammation-inducing and pro-fibrosis genes and had more liver damage when they were fed an HFD. Notably, when *Aoah*[-/-] mice were young and had not developed MASLD, their livers already expressed significantly elevated levels of nSREBP1 and its target genes (***Figure 7***).

Liver is an important tissue that converts carbohydrates into lipids. SREBP1c, the predominant isoform expressed in the liver, plays an important role in fatty acid synthesis (***Shimano and Sato, 2017***). SREBP1c mRNA was elevated in MASLD patient livers (***Kohjima et al., 2008***) and its chronic activation contributed to MASLD progression (***Kawano and Cohen, 2013***) SREBP1 has become a target for MASLD treatment (***Ju et al., 2020***; ***Jump et al., 2013***). Intriguingly, nSREBP1 levels increased in the livers of 6–8-week-old (i.e., young) *Aoah*[-/-] mice before MASLD developed; the expression of many SREBP1 target genes, such as those involved in fatty acid biosynthesis (*Acly*, *Acaca*, *Acacb*, *Fasn*, *Scd1*, *Acss2*), also increased. In addition, *Srebf1a* and *Srebf1c* mRNA both increased in *Aoah*[-/-] mouse livers, suggesting that regulation occurs at the transcription level or, because the *Srebf1* gene (encoding SREBP1) promoter contains SREs, increased nSREBP1 induced a feed-forward transcription of SREBP1 (***DeBose-Boyd and Ye, 2018***). We found that orally gavaged LPS increased hepatic LPS, nSREBP1 abundance, and the expression of nSREBP1's target genes in *Aoah*[+/+] mice, suggesting that gut-derived LPS reaches the liver and promotes fatty acid synthesis. Thus, our data suggest that when AOAH is lacking, excessive gut-derived LPS stimulates SREBP1 activation to promote de novo lipogenesis in the liver, contributing to more severe MASLD.

Notably, AKT-mTOR-S6K activity increased in *Aoah*[-/-] mouse livers; it may contribute to increased SREBP1 translocation and processing (***DeBose-Boyd and Ye, 2018***; ***Jeon et al., 2023***; ***Owen et al., 2012***). When we isolated primary mouse hepatocytes and stimulated them with LPS in vitro, we found that nSREBP1 upregulation was induced in an mTOR-dependent manner, suggesting that LPS stimulates hepatocyte directly to promote mTOR activation, which induces lipid accumulation.

In addition to fatty acid biosynthesis gene expression, the expression of *Cd36*, which is involved in free fatty acid uptake (***Chen et al., 2022***), also increased in the livers of young *Aoah*[-/-] mice while the expression of fatty acid oxidation-related genes *Acot2* and *Ppara* decreased (***Bougarne et al., 2018***; ***Moffat et al., 2014***). In addition to taking up fatty acids, CD36 interacts with INSIG2, a negative regulator of SREBP1, promoting the translocation of SREBP1 from ER to Golgi for cleavage and activation (***Zeng et al., 2022***). In keeping with our findings, Kim et al. found that LPS suppressed PPAR-α expression via ERK activation and HNF4 phosphorylation in primary mouse hepatocyte culture (***Kim et al., 2024***). In addition to promoting hepatic FA oxidation, PPAR-α is a transactivating factor that enhances Insig2 expression in hepatocytes, preventing SREBP activation (***Lee et al., 2017***). These results suggest that hepatic LPS stimulation promotes lipid accumulation via many mechanisms. In a previous study, Huang et al. found that LPS and other TLR agonists promoted fat retention in murine macrophages by increasing TAG synthesis and reducing lipolysis, yet fatty acid synthesis gene abundance did not change (***Huang et al., 2014***). In contrast, we found that *Aoah*[-/-] and *Aoah*[+/+] mouse livers had similar levels of *Acsl1*, *Dgat2*, and *Pnpla2* mRNAs, suggesting that in response to LPS or other PAMPs, hepatocytes and macrophages may accumulate fat via different mechanisms. Notably,

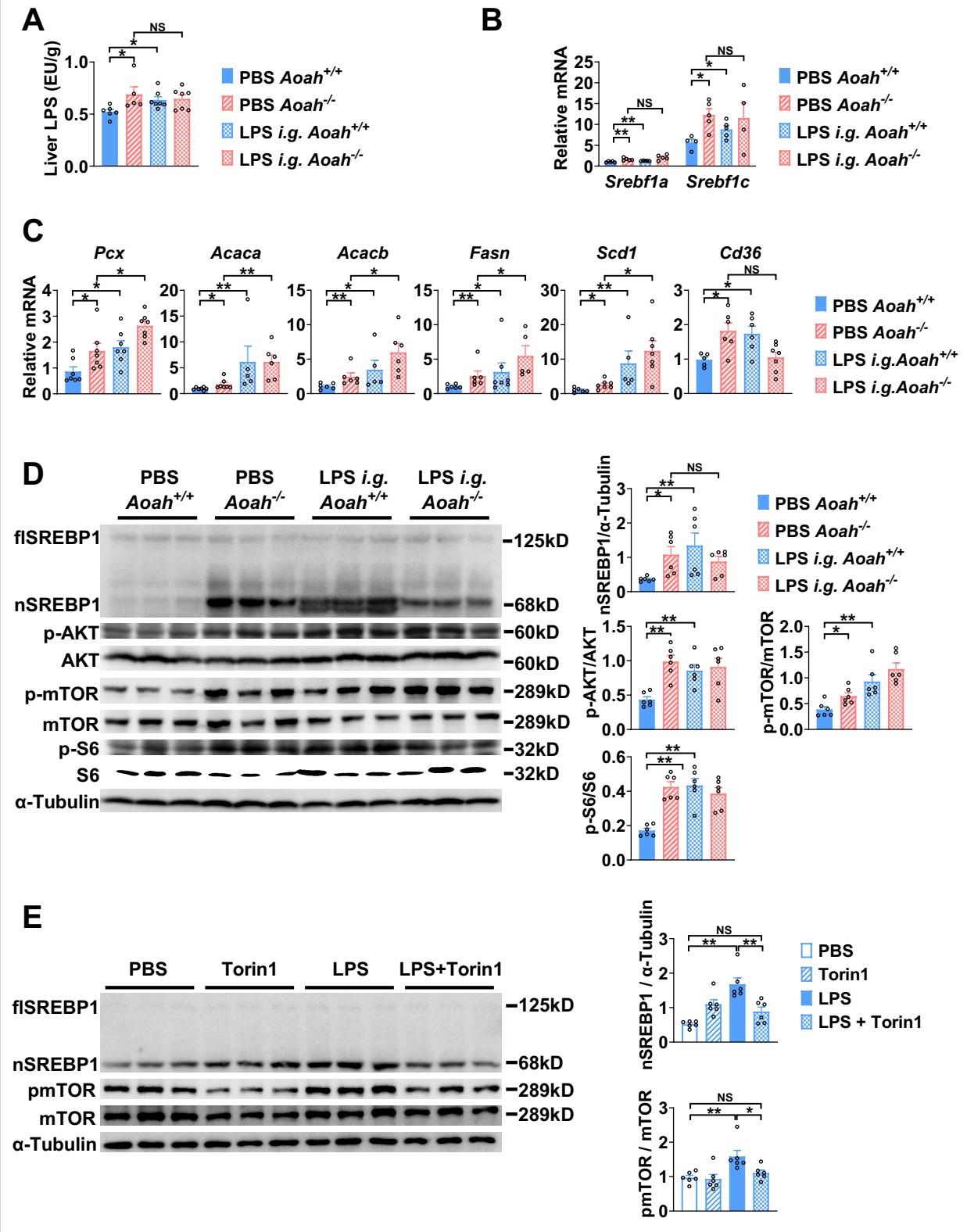

**Figure 6.** Excessive gut-derived lipopolysaccharides (LPS) increases hepatic nSREBP1 and mTOR activation. Livers from *Aoah*[+/+] mice, *Aoah*[-/-] mice, and *Aoah*[+/+] mice that were orally gavaged (i.g.) with 200 μg LPS 24 h earlier were obtained. (**A**) Hepatic LPS levels were measured. (**B**) *Srebf1a* and *Srebf1c* mRNA was measured in livers. (**C**) The mRNA levels of fatty acid biosynthesis-related genes and *Cd36* were measured using qPCR. (**A–C**) Data were combined from two experiments, n=5–7, each symbol represents one mouse. (**D**) SREBP1 protein levels and AKT-mTOR activities were measured

*Figure 6 continued on next page*

*Figure 6 continued*

using western and ImageJ. Livers from *Aoah*[-/-] mice and *Aoah*[+/+] mice that were orally gavaged (i.g.) with 200 µg LPS 24 h earlier had higher nSREBP1 and mTOR activities than did those from control *Aoah*[+/+] mice. Data were combined from two experiments, n=6, each symbol represents one mouse. (**E**) Primary hepatocytes were isolated from co-housed 6–8-week-old *Aoah*[+/+] mice and treated with 10 ng/ml LPS with or without 100 nM Torin1 for 6 h. Cells were lysed for western analysis. Data were combined from two experiments, n=6. Mann–Whitney test was used. *p<0.05; **p<0.01; ***p<0.001.

The online version of this article includes the following source data for figure 6:

**Source data 1.** Original tiff files of western blots for *Figure 6D and E*.

**Source data 2.** Original tiff files containing uncropped western blots with labeling for *Figure 6D and E*.

we found previously that after LPS stimulation of macrophages in vitro, the culture medium became acidic due to aerobic glycolysis (the Warburg effect). The acidic environment contributed more to increasing fat accumulation than did LPS stimulation (*Lu et al., 2014*). A sensitive pH indicator is needed to find out if the blood and/or extracellular fluid in the liver become acidic due to excessive LPS stimulation and if the acidity promotes hepatic fat accumulation.

In addition to LPS, AOAH also deacylates and inactivates oxidized phospholipids and lysophospholipids (*Zou et al., 2021*), Danger-associated molecular pattern (DAMP) molecules that are induced by inflammation and known to contribute to MASLD (*Sun et al., 2020*; *Wang et al., 2022*). By inactivating gut-derived PAMPs and DAMPs, AOAH may decrease hepatic fat accumulation and prevent MASLD. Increasing AOAH abundance may be a useful way to prevent and/or reduce this common disease.

## Materials and methods
### Mice

C57BL/6J *Aoah*[-/-] mice were produced at the University of Texas Southwestern Medical Center, Dallas, Texas, (*Lu et al., 2003*), transferred to the National Institutes of Health, Bethesda, Maryland, USA, and then provided to Fudan University, Shanghai, China. The mutated *Aoah* gene had been backcrossed to C57BL/6J mice for at least 10 generations. *Aoah*[+/+] and *Aoah*[-/-] male mice were housed in a specific

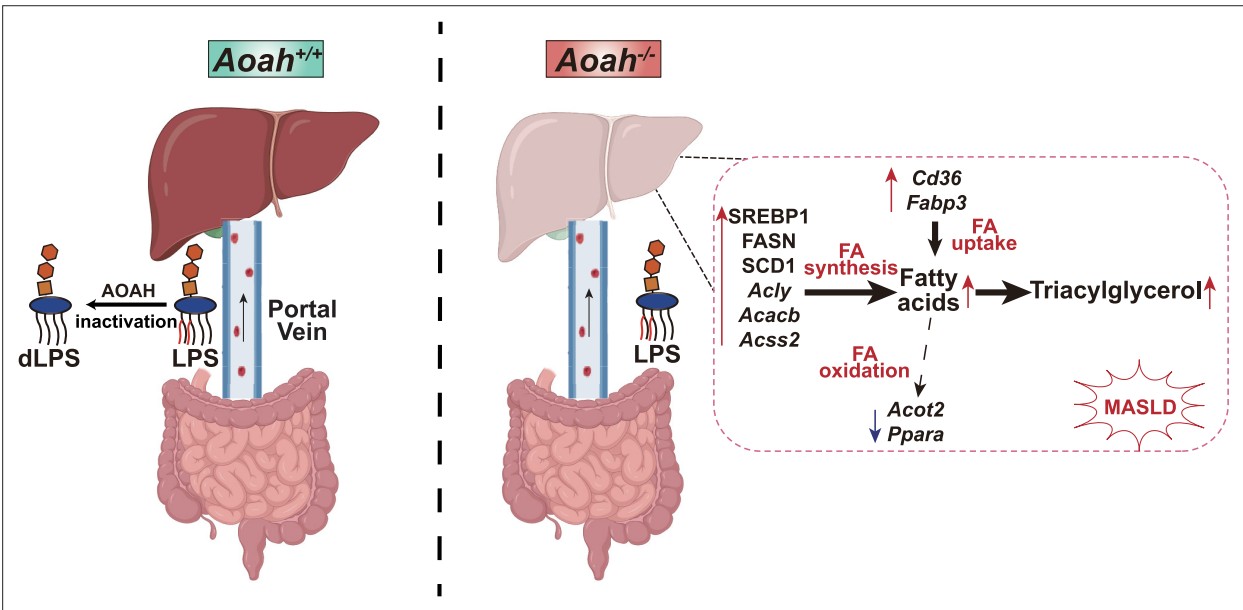

**Figure 7.** Acyloxyacyl hydrolase (AOAH) prevents metabolic dysfunction-associated steatotic liver disease (MASLD) by inactivating gut-derived lipopolysaccharides (LPS). Gut-derived LPS may translocate via the portal vein to the liver. In *Aoah*[+/+] mice, LPS can be deacylated by AOAH in the intestine and portal venous blood; when intact LPS reaches the liver, it can be inactivated by hepatic AOAH. In *Aoah*[-/-] mice, gut-derived LPS remains able to stimulate fat accumulation (steatosis) in the liver. LPS stimulates hepatocytes to generate nuclear SREBP1, which promotes fatty acid biosynthesis gene expression. LPS also increases the expression of fatty acid uptake genes *Cd36* and *Fabp3* while reducing that of fatty acid oxidation-related genes, *Acot2* and *Ppara*. Persistent LPS stimulation renders *Aoah*[-/-] mice more likely to develop MASLD than are *Aoah*[+/+] mice. dLPS = deacylated LPS.

pathogen-free facility with 12 h light/dark cycle at 22°C at the Fudan University Experimental Animal Center (Shanghai, China). Mice were randomly assigned to each experimental group. *Aoah*[+/+] and *Aoah*[-/-] male mice were co-housed for at least 3 weeks before the start and throughout the experiments. Co-housed *Aoah*[+/+] and *Aoah*[-/-] mice were processed and analyzed blindly. All studies used protocols approved by the Institutional Animal Care and Use Committee (IACUC) of Fudan University (2023-DWYY-03JZS). All animal study protocols adhered to the Guide for the Care and Use of Laboratory Animals.

## MASLD mouse model
Co-housed *Aoah*[+/+] and *Aoah*[-/-] male mice were fed either an NC or a high-fat calorie diet (D12492, Research Diets, USA) that contained protein:carbohydrate:fat (20:20:60, kcal%) plus high fructose (23.1 g/l; F3510, Sigma, USA) and glucose (18.9 g/l; G8270, Sigma) in the drinking water (HFD) for 28 weeks (*Liu et al., 2018*).

## Blood analysis
Total TAG, TCHO, LDL, HDL, AST, and ALT in mouse serum were measured at the Department of Laboratory Animal Science, Fudan University, using ADVIA Chemistry XPT.

## Liver histology
After livers were excised and fixed in 4% paraformaldehyde for 18 h, they were sectioned and stained with H&E and Oil Red O. The samples were examined for steatosis, hepatocyte ballooning degeneration, lobular inflammation, and lipid droplets using a Nikon E200 microscope.

## MASLD activity score
Histological analysis of the liver was performed based on the Sheka MASLD scoring criteria (*Sheka et al., 2020*). Liver steatosis is an infiltration of hepatic fat with minimal inflammation and is graded based on the fat percentage in hepatocytes: grade 0 (< 5%), grade 1 (5–33%), grade 2 (33–66%), and grade 3 (>66%). Inflammatory activity is manifested by two factors: grade 0 (no inflammation), grade 1 (<2 foci per 200× field), grade 2 (2–4 foci per 200× field), grade 3 (>4 foci per 200× field), and the presence of hepatocyte ballooning degeneration: no ballooned cells (grade 0), a few ballooned cells (grade 1), and many ballooned cells (grade 2).

## Liver lipid analysis
Undiluted serum samples and liver homogenates (50 mg/ml) were run in duplicate alongside a standard curve of glycerol (triglyceride assay), cholesterol (cholesterol assay), or palmitic acid (free fatty acid assay) according to the manufacturer's instructions. Triglyceride Determination Kit and Cholesterol Determination Kit were obtained from Applygen. Free Fatty Acid Quantitation Kit was purchased from Sigma-Aldrich.

## Real-time PCR (qPCR)
RNA from livers or isolated hepatocytes was purified using TRNzol Universal Reagent (Tiangen) and reversely transcribed (Tiangen). The primers used for qPCR are listed in *Supplementary file 1*. *Actb* was used as an internal control, and the relative gene expression was calculated using the ΔΔCt quantification method.

## Liver immune cell isolation
After mice were exsanguinated, 2 ml PBS containing collagenase (5 mg/ml, type IV, Sigma) were injected into the liver via the inferior vena cava. The liver was cut into small pieces and treated with collagenase (0.5 mg/ml) for 10 min at 37°C. The liver pieces were mashed using syringe plungers. The cells were passed through a 70 µm strainer (WHB scientific) and then centrifuged at 50 × *g* for 3 min, three times, to pellet hepatocytes. The immune cells in the supernatant were pelleted (500 × *g* for 15 min) and then isolated on a 40% Percoll step gradient (Cytiva). The cells were then stained and analyzed using flow cytometry (*Shao et al., 2007*).

## Flow cytometry

Liver cells were collected by centrifugation and then incubated with Fc blocking antibody (purified anti-mouse CD16/32, BioLegend, AB_2262724) on ice for 15 min. After the cells were stained with fluorescence-conjugated antibodies for 30 min on ice, they were washed and subjected to FACS (BD, FACSCelesta). The FACS data were analyzed using FlowJo software (TreeStar, Inc). All antibodies used for flow cytometry were anti-mouse antigens. Anti-mouse antibodies used for flow cytometry were anti-CD45-BV785 (Clone 30-F11, BioLegend, AB_2564590), anti-CD11b-FITC (Clone M1/70, BioLegend, AB_312788), anti-F4/80-BV421 (Clone BM8, BioLegend, AB_2563102), anti-Ly6G-PerCP-Cy5.5 (Clone 1A8, BioLegend, AB_1877271), anti-Ly6C-APC-Cy7 (Clone HK1.4, BioLegend, AB_10643867), anti-MHC II-PE-Cy7 (Clone M5/114.15.2, BioLegend, AB_2069376), anti-VSIG4-APC (Clone NLA14, eBioscience, AB_2637428), and anti-Tim4-PE (Clone RMT4-54, BioLegend, AB_1227807).

## Hepatocyte isolation

Primary hepatocytes were isolated from adult mice using a two-step collagenase perfusion method (*Charni-Natan and Goldstein, 2020*). In brief, the peritoneal cavity was opened and the liver was perfused in situ via the portal vein at 37°C with 20 ml PBS followed by 20 ml DMEM containing 10 mg collagenase (type IV, Sigma). The liver was then removed and gently minced, and the released cells were dispersed in DMEM containing 5% fetal bovine serum and 1% penicillin/streptomycin. The solution containing the mixed cells and debris was passed through a 100 µm cell strainer, and then centrifuged at 50 × *g* for 3 min, twice. The hepatocytes were pelleted and then isolated on a 90% Percoll step gradient (Cytiva). Hepatocytes were then resuspended in a TRNzol Universal Reagent (Tiangen) to measure mRNA.

## Western blot

Small pieces of liver were lysed with RIPA buffer (Biyotime) containing 1 mM PMSF (Biyotime) and a proteinase inhibitor mixture (Selleck). A commercial cytosol and nucleus Protein Extraction Kit (P0027, Beyotime) was used to separate cytosolic and nuclear proteins in the liver. The following antibodies were used for western analysis: anti-SREBP1 (SC-17755, Santa Cruz, AB_628283), anti-AOAH (HPA021666, Sigma-Aldrich, AB_3678739), anti-Lamin B1 (12987-1-AP, Proteintech, AB_2136290), anti-SCD1 (ab39969, Abcam, AB_945374), anti-FASN (SC-48357, Santa Cruz, AB_627584), anti-Phospho-mTOR (Ser2448) (5536, Cell Signaling Technology, AB_10691552), anti-Phospho-p70 S6 Ribosomal Protein (Thr389) (9234, Cell Signaling Technology, AB_2269803), anti-Phospho-AKT (Ser473) (4060, Cell Signaling Technology, AB_2315049), anti-mTOR (2983, Cell Signaling Technology, AB_2105622), anti-p70 S6 Ribosomal Protein (2217, Cell Signaling Technology, AB_331355), anti-AKT (4691, Cell Signaling Technology, AB_915783), and anti-α-Tublin (HRP-66031; Proteintech, AB_2687491). Anti-mouse IgG (7076S; Cell Signaling Technology, AB_330924) and anti-rabbit IgG (7074S, Cell Signaling Technology, AB_2099233) were used as secondary antibodies. ECL substrate (Bio-Rad Diagnostic) was used to detect proteins in western blotting, and the blot bands were quantified using ImageJ.

## Measurement of TLR4-stimulating activities in mouse feces, liver, and plasma

Fresh feces were collected and resuspended in endotoxin-free PBS (0.1 g/ml) and centrifuged at 800 × *g* for 5 min. The supernatant was heated at 70°C for 10 min. Mice were bled from the eye socket; 5 µl of 0.5 M EDTA was used as an anticoagulant. Livers were homogenized in PBS, centrifuged, and the supernatant was obtained. In some experiments, 200 µg of LPS was resuspended in 200 µl of PBS and then the suspension was slowly administered into the esophagus of mice using a gavage needle. Twenty-four hours later, the livers were collected for analysis. All the samples were collected for TLR4-stimulating activity using a cell-based colorimetric endotoxin detection kit (HEK-Blue LPS Detection Kit2, InvivoGen). Diluted samples were added to human embryonic kidney (HEK-293) cells that expressed hTLR4 and an NF-κB-inducible secreted embryonic alkaline phosphatase reporter gene. After 18 h incubation, cell culture media were applied to QUANTI-Blue medium to measure alkaline phosphatase activity. A preparation of *E. coli* 055:B5 LPS, standardized to FDA-approved control standard endotoxin, which was included in the kit, was used to quantitate TLR4-stimulating activity. Plates were read at a wavelength of 620 nm (Tecan).

## Gut permeability analysis

After mice were fasted for 18 h, they were orally gavaged with fluorescein isothiocyanate (FITC)-conjugated 4 kDa dextran (50 mg per 100 g body weight) (46944, Sigma-Aldrich). Four hours after gavage, blood was collected from the facial vein and the serum was used for FITC fluorescence measurements (excitation, 490 nm; emission, 520 nm).

## RNA-sequencing analysis

Total RNA was isolated using TRNzol from co-housed 6–8-week-old *Aoah*^(+/+) and *Aoah*^(-/-) mouse livers. The libraries were sequenced on an Ilumina Novaseq 6000 platform and 150 bp paired-end reads were generated. Differential expression analysis was performed using the DESeq2. Q value <0.05 and foldchange >1.5 were set as the threshold for significantly differential gene expression. GSEA was performed using GSEA software. RNA-seq analysis was conducted by Shanghai OE Biotech. Co, Ltd., China, and the results were deposited at PRJNA1022016.

## Single-cell RNA sequencing analysis

AOAH expression in mouse hepatic cells was analyzed based on single-cell RNA-seq data using the Liver Cell Atlas at https://www.livercellatlas.org/ (*Remmerie et al., 2020*). Liver single-cell RNA sequencing data from MASLD-cirrhosis patients (n=5) and healthy controls (n=5) were obtained from GSE136103 (*Ramachandran et al., 2019*). R package Seurat (version 5.1.0) was utilized for clustering and cell-type identification.

## Kupffer cell deletion

Kupffer cells were depleted by injecting 200 μl Clodronate-liposomes (5 mg/ml, Liposoma BV) i.v. and PBS-liposomes were used as controls. Two days after injection, Kupffer cell deletion was confirmed by staining F4/80^+ cells in cryostat liver sections.

## Statistical analysis

Data are presented as mean ± SEM. Differences between groups were analyzed using Mann–Whitney test. To compare kinetic difference, two-way ANOVA was used. The statistical significance was set at $p<0.05$. *$p<0.05$, **$p<0.01$, and ***$p<0.001$.

# Acknowledgements

This work was supported by the National Natural Science Foundation of China (32370979, 32170929, 91742104, 31770993, and 31570910 to ML, 32260189, 32300772 to WJ, 82301980 to BZ), the Shanghai Committee of Science and Technology (21ZR1405400 to ML), Guizhou Provincial Science and Technology Projects (ZK2024-240 to WJ), Science and Technology Program of the Health Commission of Guizhou Province (gzwkj2023-568 to WJ), and National Institutes of Health (AI18188 and AI44642 to RM).

# Additional information

### Competing interests

Benkun Zou: affiliated with BeiGene (Shanghai) Research & Development Co., Ltd. The other authors declare that no competing interests exist.

### Funding

| Funder | Grant reference number | Author |
|---|---|---|
| National Natural Science Foundation of China | 32370979 | Mingfang Lu |
| National Natural Science Foundation of China | 32170929 | Mingfang Lu |

| Funder | Grant reference number | Author |
|---|---|---|
| National Natural Science Foundation of China | 91742104 | Mingfang Lu |
| National Natural Science Foundation of China | 31770993 | Mingfang Lu |
| National Natural Science Foundation of China | 31570910 | Mingfang Lu |
| National Natural Science Foundation of China | 32260189 | Wei Jiang |
| National Natural Science Foundation of China | 32300772 | Wei Jiang |
| National Natural Science Foundation of China | 82301980 | Benkun Zou |
| Shanghai Committee of Science and Technology | 21ZR1405400 | Mingfang Lu |
| Guizhou Provincial Science and Technology Department | ZK2024 - 240 | Wei Jiang |
| Science and Technology Program of Guizhou Province | gzwkj2023-568 | Wei Jiang |
| National Institutes of Health | AI18188 | Robert S Munford |
| National Institutes of Health | AI44642 | Robert S Munford |

The funders had no role in study design, data collection and interpretation, or the decision to submit the work for publication.

## Author contributions

Zhiyan Wang, Data curation, Formal analysis, Validation, Investigation, Visualization, Writing – original draft; Nore Ojogun, Investigation, Methodology; Yiling Liu, Lu Gan, Zeling Xiao, Jintao Feng, Yeying Chen, ChengYun Yu, Changshun Li, Asha Ashuo, Investigation; Wei Jiang, Benkun Zou, Funding acquisition, Investigation; Xiaobo Li, Mingsheng Fu, Jian Wu, Yiwei Chu, Resources, Methodology; Robert S Munford, Conceptualization, Resources, Funding acquisition, Methodology, Writing – review and editing; Mingfang Lu, Conceptualization, Data curation, Formal analysis, Supervision, Funding acquisition, Validation, Methodology, Writing – original draft, Project administration, Writing – review and editing

## Author ORCIDs

Robert S Munford  https://orcid.org/0000-0003-1509-1294
Mingfang Lu  https://orcid.org/0000-0002-8612-3444

## Ethics

All studies used protocols approved by the Institutional Animal Care and Use Committee (IACUC) of Fudan University (2023-DWYY-03JZS). All animal study protocols adhered to the Guide for the Care and Use of Laboratory Animals.

Reviewer #1 (Public review): https://doi.org/10.7554/eLife.100731.3.sa1
Reviewer #2 (Public review): https://doi.org/10.7554/eLife.100731.3.sa2
Author response https://doi.org/10.7554/eLife.100731.3.sa3

# Additional files

## Supplementary files

MDAR checklist

Supplementary file 1. Primers used for qPCR.

## Data availability

The RNA-seq data were deposited at PRJNA1022016.

The following dataset was generated:

| Author(s) | Year | Dataset title | Dataset URL | Database and Identifier |
|---|---|---|---|---|
| Wang Z | 2023 | Mouse liver RNA-seq | https://www.ncbi.nlm. nih.gov/bioproject/ PRJNA1022016 | NCBI BioProject, PRJNA1022016 |

The following previously published datasets were used:

| Author(s) | Year | Dataset title | Dataset URL | Database and Identifier |
|---|---|---|---|---|
| Scott CL, Guilliams M | 2022 | Spatial proteogenomics reveals distinct and evolutionarily-conserved hepatic macrophage niches | https://www.ncbi. nlm.nih.gov/geo/ query/acc.cgi?acc= GSE192742 | NCBI Gene Expression Omnibus, GSE192742 |
| Ramachandran P, Henderson NC, Wilson-Kanamori JR | 2019 | Resolving the fibrotic niche of human liver cirrhosis using single-cell transcriptomics | https://www.ncbi. nlm.nih.gov/geo/ query/acc.cgi?acc= GSE136103 | NCBI Gene Expression Omnibus, GSE136103 |

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
