## [Editor Report · eLife Assessment]

This **important** study highlights the key role of the gut–liver axis mediated by LPS in causing hepatic steatosis. The authors provide **solid** evidence, in vivo, in vitro, and in silico, for the role of acyloxyacyl hydrolase in mediating this effect using KO mice subjected to MASD-inducing diets. The findings are significant for the liver research community and others interested in the gut–liver axis.

---

## [Referee Report · Reviewer #1 (Public review)]

Lu et. al. proposed here a direct role of LPS in inducing hepatic fat accumulation and that metabolism of LPS therefore can mitigate fatty liver injury. With an Acyloxyacyl hydrolase whole-body KO mice, they demonstrated that Acyloxyacyl hydrolase deletion resulted in higher hepatic fat accumulation over 7 months of high glucose/high fructose diet. Previous literature has found that hepatocyte TLR4 (which is a main receptor for binding LPS) KO reduced fatty liver in MAFLD model, and this paper complement this by showing that degradation/metabolism of LPS can also reduce fatty liver. Using clodronate-liposomes to deplete KC, the authors went on to show that AOAH level decreased significantly with increased SREBP1 level, suggesting that KCs were the major source of AOAH in the liver. To explain the mechanism of LPS induced lipogenesis, the authors demostrated in vitro that LPS alone without KC can induce SREBP1 level in primary hepatocytes via mTOR activation. This result proposed a very interesting mechanism, and the translational implications of utilizing Acyloxyacyl hydrolase to decrease LPS exposure is intriguing.

The strengths of the present study include that they raised a very simplistic mechanism with LPS that is of interest in many diseases. The phenotype shown in the study is strong. The mechanism proposed by the findings are generally well supported. Manuscript significantly improved with revision. Overall, this work adds to the current understanding of the gut-liver axis and development of MAFLD, and will be of interest to many readers.

---

## [Referee Report · Reviewer #2 (Public review)]

The authors of this article investigated the impact of the host enzyme AOAH on the progression of MASLD in mice. To achieve this, they utilized whole-body Aoah-/- mice. The authors demonstrated that AOAH reduced LPS-induced lipid accumulation in the liver, probably by decreasing the expression and activation of SREBP1. In addition, AOAH reduced hepatic inflammation and minimized tissue damage.

The authors have effectively addressed some key questions I raised. However, I still have some lingering concerns regarding the mechanisms underlying AOAH's effects.

(1) AOAH is expressed in the intestine, where it may inactivate LPS before it enters systemic circulation. In Fig. 3F, fecal LPS is significantly higher in Aoah⁻/⁻ mice compared to Aoah⁺/⁺ mice, indicating that AOAH in the intestine reduces bioactive LPS levels at the source. This implies that differences in hepatic LPS levels are already influenced by the gut environment, raising doubts about how much Kupffer cells contribute to inactivating LPS in the liver.

(2) The reliance on Kupffer cell depletion with clodronate-liposomes may overestimate the role of Kupffer cells because clodronate does not exclusively target hepatic Kupffer cells. Clodronate liposomes are taken up by macrophages systemically, potentially depleting macrophages in other organs, including the intestine and circulation. This means observed effects could also be due to loss of AOAH activity in non-hepatic macrophages.

---

## [Author Response]

The following is the authors’ response to the original reviews

**Public Reviews:**

**Reviewer #1 (Public review):**
Lu et. al. proposed here a direct role of LPS in inducing hepatic fat accumulation and that the metabolism of LPS therefore can mitigate fatty liver injury. With an Acyloxyacyl hydrolase whole-body KO mice, they demonstrated that Acyloxyacyl hydrolase deletion resulted in higher hepatic fat accumulation over 8 months of high glucose/high fructose diet. Previous literature has found that hepatocyte TLR4 (which is a main receptor for binding LPS) KO reduced fatty liver in the MAFLD model, and this paper complements this by showing that degradation/metabolism of LPS can also reduce fatty liver. This result proposed a very interesting mechanism and the translational implications of utilizing Acyloxyacyl hydrolase to decrease LPS exposure are intriguing.The strengths of the present study include that they raised a very simplistic mechanism with LPS that is of interest in many diseases. The phenotype shown in the study is strong. The mechanism proposed by the findings is generally well supported.There are also several shortcomings in the findings of this study. As AOAH is a whole-body KO, the source production of AOAH in MAFLD is unclear. Although the authors used published single-cell RNA-seq data and flow-isolated liver cells, physiologically LPS degradation could occur in the blood or the liver. The authors linked LPS to hepatocyte fatty acid oxidation via SREBP1. The mechanism is not explored in great depth. Is this signaling TLR4? In this model, LPS could activate macrophages and mediate the worsening of hepatocyte fatty liver injury via the paracrine effect instead of directly signaling to hepatocytes, thus it is not clear that this is a strictly hepatocyte LPS effect. It would also be very interesting to see if administration of the AOAH enzyme orally could mitigate MAFLD injury. Overall, this work will add to the current understanding of the gut-liver axis and development of MAFLD and will be of interest to many readers.

We thank the reviewers for their important questions and comments.

In previous studies we found that AOAH is expressed in Kupffer cells and dendritic cells cells (Shao et al., 2007). Single-cell RNAseq analysis of mouse livers by others has found AOAH in Kupffer cells, monocytes, NK cells and ILC1 cells (Remmerie et al.,2020). We also analyzed human liver single-cell RNAseq data and found that AOAH is expressed in monocytes, macrophages, resident and circulating NK cells, and some T cells (Ramachandran et al., 2019) (Please see new Figure 3E). Using clodronate-liposomes to deplete Kupffer cells we found that hepatic AOAH mRNA diminished and nSREBP1 increased (Please see new Figure 5D). These results suggest that Kupffer cells are the major source of AOAH in the liver and that LPS needs to be inactivated in the liver to prevent hepatocyte lipid accumulation.

Using primary hepatocyte culture, we found that LPS can stimulate hepatocytes directly to induce mTOR activation and SREBP1 activation (new Figure 6E). Adding purified Kupffer cells to the hepatocyte culture did not further increase SREBP1 activation. These results suggest that LPS may directly stimulate hepatocyte to accumulate fat, at least in vitro.

Both TLR4 and caspase 11 are reported to play important roles in MASLD development (Sharifnia et al., 2015; Zhu et al., 2021). We have crossed Aoah^-/-^ mice with TLR4^-/-^ mice and found that Aoah^-/-^TLR4^-/-^ and Aoah^-/-^ mice had similarly severe MASLD. This is probably because TLR4 is required for gut homeostasis (Rakoff-Nahoum et al., 2004); in TLR4 whole-body KO mice compromised gut homeostasis may result in more severe MASLD. By specifically deleting TLR4 on hepatocytes, Yu et al found that NASH-induced fibrosis was mitigated (Yu et al., 2021). In future studies we therefore would need to specifically delete TLR4 in hepatocytes to test whether excessive gut-derived LPS in Aoah^-/-^ mice stimulates hepatic TLR4 to induce more severe MASLD. We would also test whether Caspase 11 is required for hepatic fat accumulation in Aoah^-/-^ mice.

It is intriguing to test whether providing exogenous AOAH may mitigate MASLD. We will use an AAV expressing AOAH to test this idea.

**Reviewer #2 (Public review):**
The authors of this article investigated the impact of the host enzyme AOAH on the progression of MASLD in mice. To achieve this, they utilized whole-body Aoah^-/-^ mice. The authors demonstrated that AOAH reduced LPS-induced lipid accumulation in the liver, probably by decreasing the expression and activation of SREBP1. In addition, AOAH reduced hepatic inflammation and minimized tissue damage.However, this paper is descriptive without a clear mechanistic study. Another major limitation is the use of whole-body KO mice so the cellular source of the enzyme remains undefined. Moreover, since LPS-mediated SREBP1 regulation or LPS-mediated MASLD progression is already documented, the role of AOAH in SREBP1-dependent lipid accumulation and MASLD progression is largely expected.Specific comments:(1) The overall human relevance of the current study remains unclear.

It is a good point. We have studied human relevance and show the results in Figure 3E. AOAH expression increased in the hepatic macrophages and monocytes of MASLD patients.

(2) Is AOAH secreted from macrophages or other immune cells? Are there any other functions of AOAH within the cells?

AOAH can be secreted from kidney proximal tubule cells and the released AOAH can be taken up by cells that do not express AOAH (Feulner et al., 2004). AOAH can also deacylate oxidized phospholipids, DAMP molecules (Zou et al., 2021).

(3) Due to using whole-body KO mice, the role of AOAH in specific cell types was unclear in this study, which is one of the major limitations of this study. The authors should at least conduct in vitro experiments using a co-culture system of hepatocytes and Kupffer cells (or other immune cells) isolated from WT or Aoah^-/-^ mice.

Thanks for the suggestion.

Using clodronate-liposomes, we depleted Kupffer cells and found that hepatic AOAH mRNA diminished and nSREBP1 increased in the liver (Please see new Figure 5D). These results confirm that Kupffer cells are the major source of AOAH in the liver and LPS needs to be inactivated in the liver to prevent hepatocyte lipid accumulation. Using primary hepatocyte culture, we found that LPS can stimulate hepatocytes directly to induce mTOR activation and SREBP1 activation (new Figure 6E). These results suggest that LPS may directly stimulate hepatocytes to accumulate fat, at least in vitro.

(4) It has been well-known that intestinal tight junction permeability is increased by LPS or inflammatory cytokines. However, in Figure 3E, intestinal permeability is comparable between the groups in both diet groups. The authors should discuss more about this result. In addition, intestinal junctional protein should be determined by Western blot and IHC (or IF) to further confirm this finding.

We have stained ZO-1 (Please see Author response image 1, ZO-1- green fluorescence) in Aoah^+/+^ and Aoah^-/-^ mouse colonic sections. We did not see a big difference between the two strains of mice.

Feeding a high fat diet in our mouse facility for 28 weeks has led to increased gut permeability, but there was no difference between Aoah^+/+^ and Aoah^-/-^mice. Thus, the more severe MASLD in Aoah^-/-^ mice is mainly caused by elevated bioactive LPS instead of increased LPS translocation from the intestine to the liver.

(5) In Figure 6, the LPS i.g. Aoah^-/-^ group is missing. This group should be included to better interpret the results.

Please see new Figure 6. When we orally gavaged Aoah^-/-^ mice with LPS, fecal LPS levels did not increase further. Their liver SREBP1 did not increase further while the SREBP1 target gene expression increased when compared with Aoah^-/-^ mice i.g. PBS.

(6) The term NAFLD has been suggested to be changed to MASLD as the novel nomenclature according to the guidelines of AASLD and EASL.

Thanks for the suggestion. We have changed NAFLD to MASLD.

**Recommendations for the authors:**

**Reviewer #1 (Recommendations for the authors):**
Consider using MAFLD rather than NAFLD.

Thanks for the suggestion. We have changed NAFLD to MASLD.

References

Feulner, J.A., M. Lu, J.M. Shelton, M. Zhang, J.A. Richardson, and R.S. Munford. 2004. Identification of acyloxyacyl hydrolase, a lipopolysaccharide-detoxifying enzyme, in the murine urinary tract. Infection and immunity 72:3171-3178.

Zou, B., M. Goodwin, D. Saleem, W. Jiang, J. Tang, Y. Chu, R.S. Munford, and M. Lu. 2021. A highly conserved host lipase deacylates oxidized phospholipids and ameliorates acute lung injury in mice. eLife 10: